# Multi-scale Change-Aware Transformer for Remote Sensing Image Change Detection

## ABSTRACT

Change detection identifies differences between images captured at different times. Real-world change detection faces challenges from the diverse and intricate nature of change areas, while current datasets and algorithms are often limited to simpler, uniform changes, reducing their effectiveness in practical application. Existing dual-branch methods process images independently, risking the loss of change information due to insufficient early interaction. In contrast, single-stream approaches, though improving early integration, lack efficacy in capturing complex changes. To address these issues, we introduce a novel single-stream architecture, the Multi-scale Change-Aware Transformer (MACT), which features the Dynamic Change-Aware Attention module and the Multi-scale Change-Enhanced Aggregator. The Dynamic Change-Aware Attention module, integrating local self-attention and cross-temporal attention, conducts dynamic iteration on images differences, thereby targeting feature extraction of change areas. The Multi-scale Change-Enhanced Aggregator enables the model to adapt to various scales and complex shapes through local change enhancement and multiscale aggregation strategies. To overcome the limitations of existing datasets regarding the scale diversity and morphological complexity of change areas, we construct the Mining Area Change Detection dataset. The dataset offers a diverse array of change areas that span multiple scales and exhibit complex shapes, providing a robust benchmark for change detection. Extensive experiments demonstrate that the our model outperforms existing methods, especially for irregular and multi-scale changes.

## CCS CONCEPTS

• **Computing methodologies** → *Image segmentation*.

## KEYWORDS

Change detection, benchmark dataset, single-stream framework

## 1 INTRODUCTION

Change Detection (CD) identifies changes in surface or phenomena over time by analyzing images captured at different time points. These changes range from local alterations, such as the construction or demolition of buildings [18, 24], to broader regional transformations like deforestation [23] and urban expansion [8, 25]. Additionally, change areas present irregular boundaries and diverse

*ACM MM, 2024, Melbourne, Australia*

© 2024 Copyright held by the owner/author(s). Publication rights licensed to ACM.
ACM ISBN 978-x-xxxx-xxxx-x/YY/MM
https://doi.org/10.1145/nnnnnnn.nnnnnnn

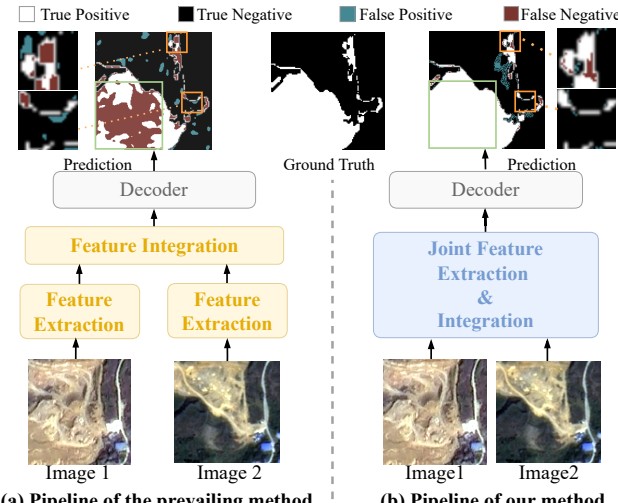

**Figure 1: Our method targets feature extraction in change areas within the encoder, aimed at precisely capturing multiscale and complex change regions.**

shapes, influenced by factors such as natural terrain, human activities and varying image resolutions. Consequently, change detection algorithms and datasets are required to capture the complex and multi-scale changes for practical monitoring. Despite providing valuable data on urban growth and agricultural changes (see Tab. 1), datasets are limited by their focus on uniform, small-scale changes. This restricts the capacity of current algorithms to accurately detect complex, multi-scale changes, reducing their practical effectiveness.

Current change detections [1, 2, 4, 5] often use dual-branch pipelines that extract and later integrate features from dual-temporal images, as shown in Fig. 1(a). These methods extract features from image pairs individually but fail to integrate early cross-temporal changes effectively, impacting accurate detection and increasing model complexity. Alternatively, a few algorithms use a single-stream approach by merging image pairs for simultaneous processing, which enhances the complementarity of information between images. However, single-stream methods, though effective for regular changes, often struggle with capturing details and adapting to multi-scale changes, particularly in complex change scenarios.

In response to the limitations faced by current change detection methods in capturing complex variations, we expect to facilitate interaction between image pairs at the early stage of image processing to refine feature extraction within change areas, as illustrated in Fig. 1(b). Additionally, we are committed to enhancing the model's ability to recognize multi-scale and complex change areas, thereby improving its effectiveness in practical applications.

Therefore, we introduce an single-stream architecture termed the Multi-scale Change-Aware Transformer (MACT). The framework adopts an encoder-decoder structure, with the Change-Aware Encoder composed of a stack of Dynamic Change-Aware Attention (DCAA) modules. The Dynamic Change-Aware Attention (DCAA) mechanism is designed to facilitate dynamic image comparison across disparate time points by harnessing the power of local self-attention and cross-temporal attention. The local self-attention mechanism extracts features from individual images and the cross-temporal attention compares the changes between two images. Furthermore, the Dynamic Change-Aware Attention adjust the size of local windows at different stages, facilitating a comprehensive and precise capture of variations across various scales. This early-stage deep feature interaction enables the encoder to optimize the recognition of features within change regions, thereby enhancing overall change detection performance.

To enhance the Multi-scale Change-Aware Transformer's capacity to detect changes across various scales, we integrate the Multis-cale Change-Enhanced Aggregator (MCEA). MCEA consists of two complementary sub-modules: the Change-Enhanced Module (CEM) and the Multi-scale Change Aggregator (MCA). The Change-Enhanced Module employs a local attention to emphasize emphasize regions within the image that exhibit significant changes. Through Depthwise Convolution, it refines the local details of features. Subsequently, the Multi-scale Change Aggregator integrates the multi-scale features output by the Change-Enhanced Module across different levels. Through a hierarchical processing strategy, Multi-scale Change Aggregator preserves local details and promotes the integration of high-level semantic information, enhancing the model's overall understanding of complex change.

To overcome the limitation of existing change detection datasets, which focus on changes with regular shapes and limited scales, we introduce the Mining Area Change Detection (MACD) dataset. MACD covers a broader range of multi-scale change areas, including irregular shapes and intricate boundary conditions, aligning closely with the complexities encountered in real-world monitoring scenarios. The Mining Area Change Detection dataset not only enriches the benchmark data for change detection but also provides a more challenging test platform, particularly in improving the detection accuracy of irregular changes.

Extensive experiments demonstrate that the Multi-scale Change-Aware Transformer has achieved state-of-the-art performance, particularly in handling complex and multi-scale change scenarios. Our contributions can be summarized as follows:

- We introduce Multi-scale Change-Aware Transformer, a novel framework leveraging Dynamic Change-Aware Attention for efficient and parallel feature extraction, with a focus on capturing features within change regions.
- We develop Multi-scale Change-Enhanced Aggregator, a component augmenting the model's ability to perceive changes at multiple scales by aggregating and refining features.
- We construct the Mining Area Change Detection dataset, which introduces complex and varied change area morphologies, providing a rich and challenging benchmark for the field of change detection.

## 2 RELATED WORK

**Change Detection.** Currently, the majority of change detection algorithms [10, 20, 27, 29] adopt a dual-branch architecture. This architecture involves independent feature extraction from dual-temporal images, fusion of these features, and prediction of change maps. Early methods [6, 16] utilized fully convolutional neural networks for feature extraction, followed by simple operations like addition or subtraction for feature fusion. In modern change detection frameworks, mechanisms such as spatial and channel attention are incorporated to enhance performance. For example, DASNet [5] combines these attention mechanisms, while DTCDSCN [17] leverages dual attention modules to exploit feature interdependencies. However, dual-branch approaches have limitations in fully integrating complementary information between dual-temporal images and often involve complex models with numerous parameters. Although single-stream structures simplify model design and facilitate interaction between image pairs, they still fall short in capturing local details and adapting to multi-scale changes, especially in complex change scenarios. This study builds upon the single-stream structure by enabling interaction between image pairs, thereby enhancing the model's flexibility and adaptability in complex scenes.

**Transformer-Based Methods for Change Detection.** Transformers, renowned for their robustness and high performance, are increasingly utilized in various change detection methods. For instance, MSTDSNet-CD [22] and SwinSUNet [26] employ dual-stream networks for feature extraction and multi-scale aggregation, albeit with high computational demands. Conversely, BIT [3] proposes a lightweight model featuring a single-stream transformer and two decoders for efficient spatiotemporal context modeling. ChangeFormer [1] combines hierarchical transformer encoders and multi-layer perceptron decoders in a Siamese network structure. Despite their commendable performance, these methods predominantly rely on dual-stream architectures. In this paper, we introduce a novel perspective by embracing a single-stream transformer.

**Datasets for Change Detection.** The mainstream change detection datasets are summarized in Tab. 1. These datasets can be roughly categorized into three types: the first category includes datasets focusing on building changes, such as LEVIR [4], WHU [12], and ABCD [11], as well as those involving land cover changes like ZY3 [28] and OSCD [7]. The second category comprises datasets containing multiple types of change targets, such as CCD [14] and SYSU [21]. However, these datasets often only contain small-scale and regularly shaped change areas, posing significant obstacles when dealing with complex change regions in practical applications. To overcome these limitations, we propose a new dataset-Mining Area Change Detection. This dataset encompasses diverse and morphologically complex change regions, presenting new challenges and higher difficulty levels for change detection tasks.

## 3 METHOD

Given a pair of input dual-temporal images $I_1$ and $I_2$, each with a shape of H × W × 3,our goal is to learn a mapping function $\Phi(\cdot)$. The $\Phi(\cdot)$ takes $I_1$ and $I_2$ as inputs and predicts the change map $P \in \mathbb{R}^{H \times W}$. Mathematically, this process can be represented as:

$$P = \Phi(I_1, I_2). \tag{1}$$

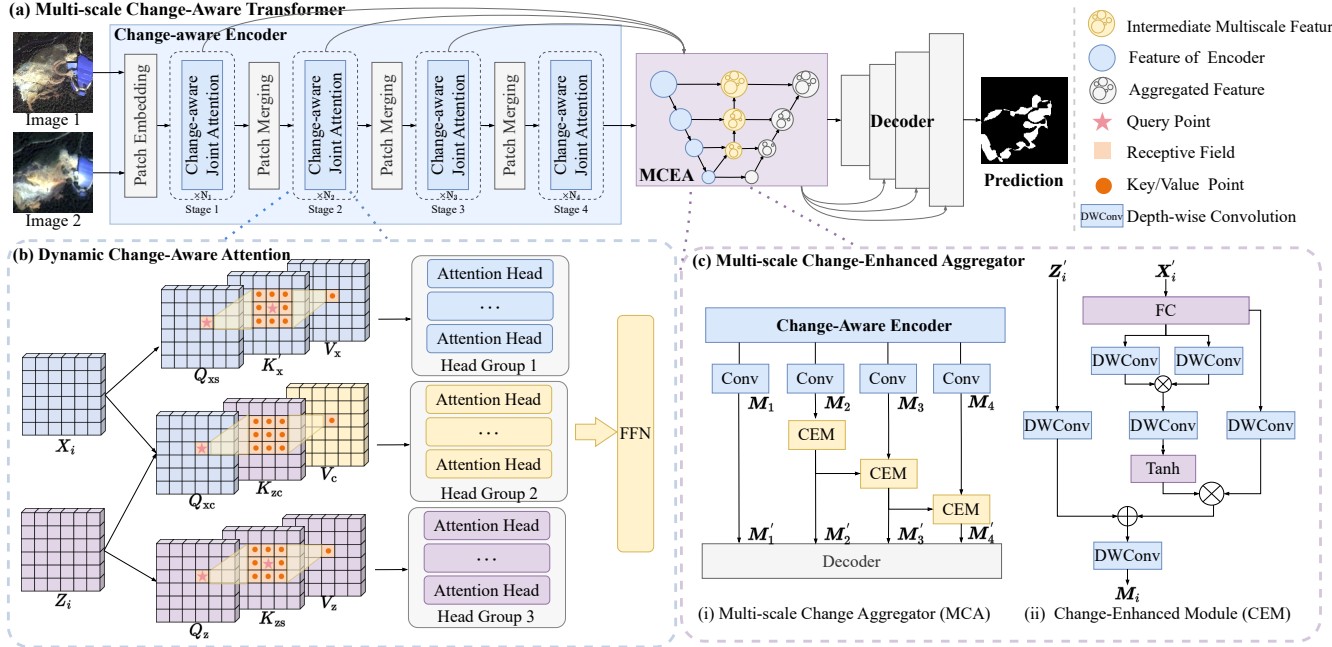

**Figure 2: (a) The core components of the Multi-scale Change-Aware Transformer are Dynamic Change-Aware Attention and Multi-scale Change-Enhanced Aggregator. (b) Details of Dynamic Change-Aware Attention. Local self-attention and cross-temporal attention are allocated to different head groups, enabling parallel feature extraction and relationship modeling. (c) Multi-scale Change-Enhanced Aggregator integrates features across scales for improved change recognition.**

To realize the mapping function $\Phi(\cdot)$, we propose the Multi-scale Change-Aware Transformer (MACT). Firstly, the Change-Aware Encoder of the MACT employs the Dynamic Change-Aware Attention to extract key features from the dual-temporal images and dynamically compare changes between image pairs. Subsequently, the extracted features are passed to the Change-Enhanced Multiscale Aggregator, which enhances sensitivity to change areas through aggregation strategies. Finally, a simple decoder transforms features into the final change map $P$, providing a detailed representation of the changes between input image pair.

## 3.1 Change-Aware Encoder

The Change-Aware Encoder is designed for feature extraction and relationship modeling of image pairs, ensuring the preservation of information about the change areas from early stages of processing.

The encoder is stacked with Dynamic Change-Aware Attention (DCAA) blocks and combines patch embedding or patch merging layers, as shown in Fig. 2(a). It consists of four stages, with each stage containing $N_i$ attention blocks, where $i$ ranges from $\{1, 2, 3, 4\}$. The images $I_1$ and $I_2$ first undergo a patch embedding layer to reduce spatial resolution and expand channel depth. Subsequently, they are processed by Dynamic Change-Aware Attention blocks across the four stages, yielding a series of multi-scale features $Y_i$.

The Dynamic Change-Aware Attention block, depicted in Fig. 2(b), serves as the cornerstone of the Change-Aware Encoder, which utilizes local self-attention to extract features and employs cross-temporal attention to contrast differences between images, thereby

facilitating change perception. The Dynamic Change-Aware Attention initiates by extracting attention elements from input images through local expansion operations. It then employs a multi-head attention to process these elements. Furthermore, Dynamic Change-Aware Attention strategically applies windows of varying sizes across different stages, facilitating a nuanced capture of changes at multiple scales.

**Gathering Attention Elements.** Formally, given dual-temporal image features $X_i$ and $Z_i$ with dimensions $H_i \times W_i \times C_i$. $H_i$ and $W_i$ represent the feature height and width at the $i$-th stage, calculated as $H_i = \frac{H}{2^{i+1}}$ and $W_i = \frac{W}{2^{i+1}}$. $C_i$ is number of channels for features. Next, the Depth-wise Separable Convolution (DWConv) is used to project features, generating queries ($Q$), keys ($K$) and values ($V$):

$$Q_x, K_x, V_x = X_i W_i^{Q_x}, X_i W_i^{K_x}, X_i W_i^{V_x}, \quad (2)$$

$$Q_z, K_z, V_z = Z_i W_i^{Q_z}, Z_i W_i^{K_z}, Z_i W_i^{V_z}, \quad (3)$$

here, $W_i^{Q_x}$, $W_i^{K_x}$, $W_i^{V_x}$, $W_i^{Q_z}$, $W_i^{K_z}$, and $W_i^{V_z}$ are the weight parameters of the convolution layer. To simplify the expression, we omit the subscript $i$ for the related representations of $Q$, $K$ and $V$.

**Local Expansion Operation.** To integrate local information and enhance sensitivity to detail changes, we perform local expansion operations on the keys $K_x$ and $K_z$. This operation effectively expands the local receptive field by considering neighborhood information around each point. Specifically, we use the **Unfold** function to extract $L_i \times L_i$ regions around the keys on the feature maps,

resulting in expanded keys $K'_x$ and $K'_z$. This process can be mathematically described as:

$$K'_x = \textbf{Unfold}(K_x, L_i), \quad (4)$$

$$K'_z = \textbf{Unfold}(K_z, L_i). \quad (5)$$

The local window size $L_i$ is adjusted according to the resolution of the feature maps at the current stage. In the early stages, a smaller window is employed to capture finer local changes, while in later stages, the window size is increased to cover a broader context, thereby detecting larger-scale changes. This dynamic window adjustment strategy enables the model to effectively capture changes at multiple scales, while optimizing the balance between computational efficiency and detection accuracy.

**Dynamic Difference Values.** In the cross-temporal attention, we devise a difference iteration process tailored to bolster the model's capability to recognize change regions. This process computes the difference between $V_x$ and $V_z$. The feature difference is computed through an absolute value operation, as depicted by the equation:

$$V_c = |V_x - V_z|. \quad (6)$$

Subsequently, the $V_c$ is utilized as the value component in the cross-temporal attention mechanism, enabling the model to focus on change regions. Thus far, we obtain the representations for queries and keys as $Q_x, K'_x, K'_z$, and for values as $V_x, V_z, V_c$, with the following dimensions: $Q_x, Q_z \in \mathbb{R}^{H_i \times W_i \times C_i \times 1}$, $K'_x, K'_z \in \mathbb{R}^{H_i \times W_i \times C_i \times L_i}$ and $V_x, V_z, V_c \in \mathbb{R}^{H_i \times W_i \times C_i \times 1}$.

**Calculating Change-Aware Attention.** As illustrated in Fig. 2(b), we partition the attention heads into three groups, with two groups dedicated to extracting features from the images at two time points using local self-attention mechanisms, while the third group employs cross-temporal attention mechanisms specifically focused on the features of change areas. Specifically, we segment the channels of the query features $Q_x$ into $Q_{xs}$ and $Q_{xc}$, and perform the same operation for the key features $K_x$ to obtain $K_{zs}$ and $K_{zc}$. Within each attention head group, the multi-head attention mechanism is utilized to compute attention features as follows:

$$\tilde{Y}_x = \textbf{MHA}(Q_{xs}, K'_x, V_x), \quad (7)$$

$$\tilde{Y}_z = \textbf{MHA}(Q_z, K_{zs}, V_z), \quad (8)$$

$$\tilde{Y}_c = \textbf{MHA}(Q_{xc}, K_{zc}, V_c), \quad (9)$$

where $\textbf{MHA}(\cdot)$ denotes the multi-head attention function. Subsequently, we concatenate the outputs of all head groups along the feature dimension to obtain $\tilde{Y}$, which is then processed through a multi-layer perceptron (MLP) to generate the final features $Y$:

$$\tilde{Y}_i = \textbf{CAT}(\tilde{Y}_x, \tilde{Y}_z, \tilde{Y}_c), \quad (10)$$

$$Y_i = \textbf{MLP}(\textbf{LN}(\tilde{Y}_i)) + \tilde{Y}_i, \quad (11)$$

where $\textbf{LN}(\cdot)$ represents the layer normalization function to accelerate model convergence. By employing three groups of attention, the model accomplishes both local self-attention and cross-temporal attention, facilitating feature extraction within the images while also focusing on features relevant to change areas, thus minimizing the loss of change information during the feature extraction stage.

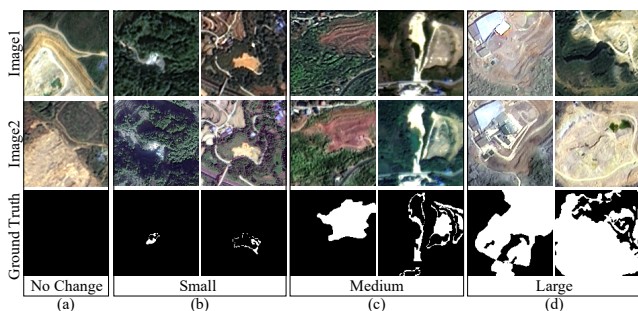

**Figure 3: Example samples from the Mining Area Change Detection dataset. The change areas exhibit a wide range of scales and intricate and variable shapes.**

## 3.2 Multi-scale Change-Enhanced Aggregator

We design a Multi-scale Change-Enhanced Aggregator (MCEA) to improve the network's feature representation of change areas. The Change-Enhanced Module (CEM) enhances local change features. Subsequently, the Multi-scale Change Aggregator (MCA) integrates the multi-scale features output by the CEM across different levels.

**Change-Enhanced Module.** To enhance the representation of local features in change areas, we introduce the Change-Enhanced Module, which combines the mechanisms of local attention and Depth-wise Separable Convolution to strengthen feature representation and highlight change areas.

Specifically, the output features $Y_i$ at each stage are decomposed into two parts along the feature dimension: $X'$ and $Z'$. $X'$ undergoes a linear mapping $f_m$ followed by two DWConv layers to aggregate local information and generate $Q_m$ and $K_m$, while $V_m$ is directly obtained from $X'$ using another DWConv layer:

$$Q_m = \textbf{DWConv}(f_m X'), \quad (12)$$

$$K_m = \textbf{DWConv}(f_m X'), \quad (13)$$

$$V_m = \textbf{DWConv}(X'), \quad (14)$$

where $f_m$ is implemented by a fully connected layer. Next, we compute the Hadamard product of $Q_m$ and $K_m$, followed by aggregation of information through a DWConv layer to generate context-aware weights $A_m$:

$$A_m = \delta(\textbf{DWConv}(Q_m \odot K_m)). \quad (15)$$

Here, $\delta(\cdot)$ is the activation function tanh introducing nonlinearity to capture more complex feature relationships. Then, these weights are combined with $V_m$ and passed through a *softmax* function $\sigma(\cdot)$ to generate the final enhanced features $M'$, which are then fused with the input feature $Z'$:

$$M' = \sigma(\frac{A_m}{\sqrt{d_m}})V_m + Z'. \quad (16)$$

Through this combination, the Change-Enhanced Module effectively emphasizes local features related to changes, reducing interference from non-change areas on the decoder, thereby improving the overall performance of change detection.

**Multi-scale Change Aggregator.** To fully exploit the potential of multi-scale features and capture complementary information across

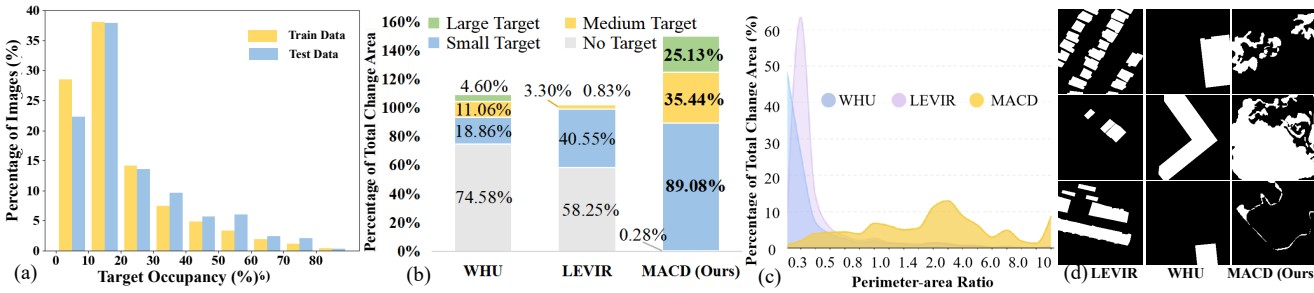

Figure 4: Statistics of the Mining Area Change Detection dataset. (a) Pixel ratio distribution in train and test data. (b) Percentage of change area for four sizes. (c) Comparison of change maps for the datasets. (d) Distribution of perimeter-area ratios to characterize the complexity of the area of change.

different levels, we propose a Multi-scale Change Aggregator. This aggregator operates in a bottom-up manner based on the enhanced features $M'$ generated by the Change-Enhanced Module.

Specifically, the fused features $M'_i$ from lower levels are not only forwarded to the corresponding decoder levels but also propagated upwards to fuse with shallower features $M_{i-1}$. This process is mathematically described by the function:

$$M'_{i-1} = f_{\text{MCA}}(M'_{i-1} + M_i), \qquad (17)$$

where $i$ denotes the stage number, and $f_{\text{MCA}}(\cdot)$ is a function obtained through upsampling operations and a $3 \times 3$ convolutional kernel with learnable parameters. The upsampling operation adjusts the spatial resolution of features to match those of shallower features, while the convolutional kernel captures and integrates feature information from different levels.

Through Multi-scale Change Aggregator, the model effectively integrates features from different levels, enhancing its capability to recognize changes across various scales.

## 3.3 Loss Function

We employ the loss function set $\mathcal{L}$ to guide the model training. The overall loss is defined as follows:

$$\mathcal{L} = \mathcal{L}_{\text{p}} + \sum_{j=1}^{N} \alpha_j (\mathcal{L}_j^{\text{WBCE}} + \mathcal{L}_j^{\text{SSIM}} + \mathcal{L}_j^{\text{SIoU}}). \qquad (18)$$

Here, $\mathcal{L}_{\text{p}}$ is cross entropy loss that evaluates the accuracy of the final change map. In our model, indexed by $j$ over $N = 4$ stages, the loss function integrates Weighted Binary Cross-Entropy (WBCE) for class balance, Structural Similarity Index (SSIM) for structural coherence, and Structural Intersection over Union (SIoU) for boundary precision, optimized by stage-specific weights $\alpha_j$.

## 4 MINING AREA CHANGE DETECTION DATASET

We introduce the Mining Area Change Detection (MACD) dataset, which is the first compilation specifically tailored for change detection in mining areas. Distinguished from urban construction and land cover monitoring datasets in Tab. 1, the MACD dataset encompasses a broad range of scale variations and intricate change

Table 1: Summary of popular change detection datasets.

| Name | Object | Scale Variation | Complex Shape |
|---|---|---|---|
| WHU [12] | Building | ✓ | ✗ |
| LEVIR [4] | Buildings | ✗ | ✗ |
| ABCD [11] | Buildings | ✗ | ✗ |
| ZY3 [28] | Land Cover | ✗ | ✓ |
| CCD [14] | Complex Scenarios | ✗ | ✗ |
| SYSU [21] | Complex Scenarios | ✓ | ✗ |
| MACD | Mining Area | ✓ | ✓ |

patterns, which is designed to enhance the performance of change detection algorithms in addressing a diversity of real-world issues.

### 4.1 Construction of Dataset

**Data Collection.** Our dataset is collected from various open-pit mining areas in the Chongqing region of China. These images are sourced from four Chinese satellites: Gaofen-1, Gaofen-2, Gaofen-6, and Ziyuan-3, with resolutions ranging from 2m to 0.8m. The imagery is acquired over a period from 2018 to August 2023, encompassing a variety of weather conditions, random variations in solar elevation angles, seasonal changes, differences in illumination, and sensor variations. These broad temporal scope and diverse conditions ensure the diversity and complexity of the dataset.

**Data Annotation.** The dataset annotation is completed by experts with extensive experience in remote sensing image interpretation and profound understanding of mining configurations. The annotation process involved coarse localization, image registration, cropping, fine-grained annotation and generation of change maps. Detailed descriptions can be found in *Supplementary Materials*.

### 4.2 Analysis of Dataset

Our dataset comprises a total of 2133 pairs of temporal images, partitioned into a training set with 1801 image pairs and a test set with 332 image pairs, all with a resolution of $128 \times 128$ pixels. We conduct a detailed statistical analysis to highlight the dataset's unique characteristics compared to other datasets.

**Scale Variation.** Fig. 4(a) illustrates the distribution of change area pixel ratios in both the training and testing sets of our dataset. The

**Table 2: Quantitative Comparison on Mining Area Change Detection dataset. All scores are described in percentages (%).**

| Methods | F1 | IoU | Pre. | Rec. | OA | FLOPs | #Param. |
|---|---|---|---|---|---|---|---|
| FC-EF [6] | 62.25 | 45.19 | 71.17 | 55.32 | 82.66 | 0.89G | 1.35M |
| Siam-Conc [6] | 47.21 | 30.90 | 42.40 | 53.26 | 84.80 | 1.33G | 1.55M |
| Siam-Diff [16] | 59.03 | 41.87 | 63.16 | 55.40 | 80.96 | 1.18G | 1.35M |
| IFNet [19] | 63.28 | 46.28 | 67.04 | 59.92 | 84.47 | 27.35G | 50.71M |
| DASNet [5] | 50.04 | 33.37 | 46.87 | 53.68 | 84.80 | 6.56G | 11.33M |
| DTCDSCN [17] | 62.16 | 45.09 | 68.08 | 57.18 | 84.73 | 41.07G | 20.44M |
| SUNet [9] | 62.82 | 45.80 | 74.30 | 54.42 | 85.40 | 8.42G | 47.62M |
| BIT [3] | 65.39 | 48.57 | 71.37 | 60.33 | 85.48 | 3.31G | 31.26M |
| ChangeFormer[1] | 64.92 | 48.06 | 68.92 | 61.36 | 83.18 | 7.82G | 43.99M |
| SARAS (V2) [2] | 66.93 | 50.30 | 72.25 | 62.34 | 83.46 | 66.64G | 102.76M |
| USSFC-Net [15] | 65.34 | 48.53 | 71.62 | 60.08 | 83.18 | 1.22G | 1.52M |
| MACT (Ours) | **67.22** | **50.63** | 70.68 | 64.09 | 85.11 | 10.64G | 34.9M |

statistical graph reveals that the dataset significantly showcases a broad range of scale variations, covering change area pixel ratios from 10% to over 80%. Changes are categorized into (less than 5%), medium (5% to 15%), and large (greater than 15%) areas relative to the image size, as illustrated in Fig. 4(b). The LEVIR [4] and WHU [12] datasets predominantly contain small change areas, while large changes are relatively rare. Our dataset, on the other hand, exhibits a balanced distribution across small, medium, and large change areas, a design intended to test the performance of change detection algorithms on different scales.

**Irregular Morphology.** We use the ratio of the perimeter to area of change regions as a complexity metric, where a higher value indicates greater irregularity. As shown in Fig. 4(c), the distribution of change region complexity across various datasets reveals that our mining area dataset contains regions with higher morphological complexity compared to urban structures or land cover datasets. Similarly, Fig. 4(d) confirms the morphological complexity of mining area changes, characterized by irregular features such as fractures, overlaps, and voids. Therefore, the dataset requires algorithms to accurately identify complex boundaries and morphologies.

## 5 EXPERIMENTS

### 5.1 Implementation Details

We conduct experiments on our MACD, LEVIR [4] and WHU [12] dataset. The model is trained with the Adam optimizer [13] and exponential learning rate decay. The initial learning rate is set to 0.004, and the training process consists of 100 epochs with a batch size of 64. The comparative methods, evaluation metrics and more implementation details are detailed in the *Supplementary Materials*.

### 5.2 Experiments on MACD Dataset

**Quantitative Results.** Upon reviewing the data in Tab. 2, while SUNet achieves the highest accuracy at 74.30%, it exhibits variance in other critical metrics, indicating a possible inclination towards false positives. Comparatively, SARAS, with its scale sensitivity, delivers a more consistent and superior performance across the board. Our proposed method not only outshines SARAS but also

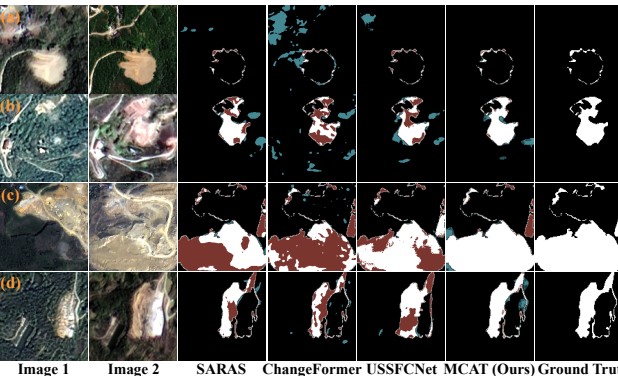

**Figure 5: Comparison of state-of-the-art change detection methods on MACD dataset. Predicted results are color-coded: white for true positives, black for true negatives, green for false positives, and red for false negatives.**

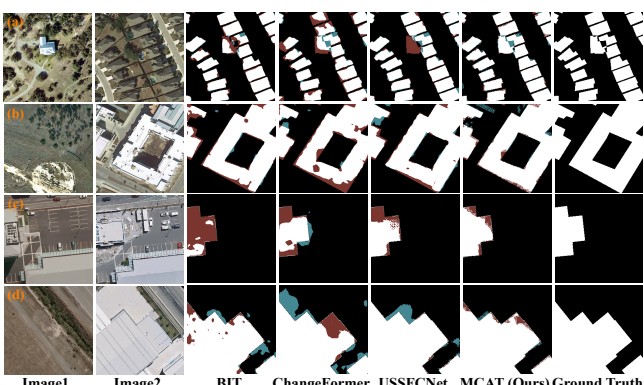

**Figure 6: Comparison results on LEVIR (examples in (a) and (b)) and WHU (examples in (c) and (d)) datasets.**

significantly improves upon it, particularly with a 1.75% increase in recall—a testament to the effectiveness of our early feature interaction approach for pinpointing changes. Moreover, Our Transformer demonstrates a 0.29% enhancement in F1 score and a 0.33% uptick in IoU, underscoring its robustness and precision in change detection tasks. These quantitative leaps highlight our method's advanced capability in handling the nuanced complexities of change detection. **Efficiency Analysis.** From Tab. 2, Multi-scale Change-Aware Transformer,with 10.64M parameters and 34.94G FLOPS, strikes an effective balance between performance and efficiency. In contrast, models like FC-EF, FC-Siam-conc, FC-Siam-diff, and DASNet, despite their low parameter counts, underperform in terms of F1 score and IoU metrics. MACT outperforms SARAS, which has 102.7M parameters and 66.64G FLOPS, by reducing the parameter count and computational cost by 90.1% and 47.9%, respectively, without compromising performance. Although USSFC-Net is lightweight, it shows a notable performance deficit when handling complex data. This analysis underscores Multi-scale Change-Aware Transformer's superior performance and efficiency, outpacing existing models in both parameter count and computational cost.

**Table 3: Comparison results for the LEVIR [4] and WHU [12] datasets. All scores are described in percentages (%).**

| Methods | LEVIR | | | | | WHU | | | | |
|---|---|---|---|---|---|---|---|---|---|---|
| | **F1** | **IoU** | Pre. | Rec. | OA | **F1** | **IoU** | Pre. | Rec. | OA |
| FC-EF [6] | 81.05 | 68.14 | 84.88 | 77.55 | 97.99 | 72.82 | 57.26 | 77.24 | 68.88 | 97.82 |
| FC-Siam-Diff [16] | 87.87 | 78.36 | 92.39 | 83.77 | 98.83 | 91.15 | 84.52 | 94.44 | 88.08 | 98.49 |
| FC-Siam-Conc [6] | 88.44 | 79.27 | 92.12 | 85.04 | 98.88 | 92.19 | 86.26 | 93.91 | 90.54 | 98.64 |
| DTCDSCN [17] | 88.09 | 78.71 | 90.14 | 86.12 | 98.83 | 93.04 | 87.34 | 96.81 | 89.55 | 98.81 |
| SUNet [9] | 92.37 | 85.83 | 93.40 | 91.37 | 98.68 | 90.33 | 83.13 | 92.39 | 88.36 | 98.28 |
| BIT [3] | 92.54 | 86.12 | 93.75 | 91.36 | 98.62 | 93.61 | 87.13 | **97.46** | 90.06 | 99.08 |
| DASNet [5] | 93.00 | 86.92 | 93.91 | 92.11 | 98.27 | 90.10 | 83.13 | 91.89 | 88.37 | 98.28 |
| ChangeFormer [1] | 91.45 | 84.25 | 94.39 | 88.69 | 98.38 | 91.32 | 85.26 | 93.39 | 89.34 | 98.54 |
| SARAS (V2) [2] | 92.59 | 86.20 | 93.76 | 91.45 | 98.30 | 89.90 | 86.15 | 91.43 | 88.42 | 98.08 |
| USSFC-Net [15] | 93.63 | 88.02 | 95.13 | 92.17 | **98.93** | 93.13 | 87.17 | 94.38 | 91.91 | 98.62 |
| MACT (Ours) | **93.76** | **88.25** | **95.19** | **92.40** | 98.84 | **93.73** | **88.21** | 95.36 | **92.15** | **98.77** |

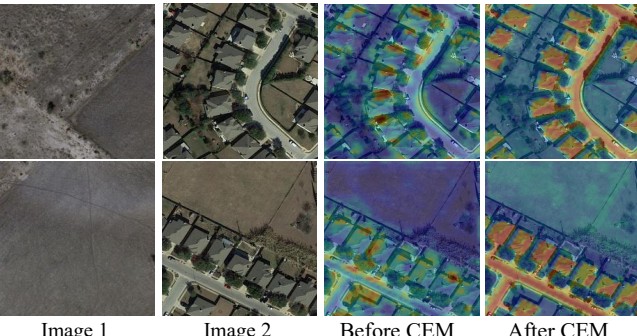

Image 1 | Image 2 | Before CEM | After CEM

**Figure 7: Comparison of feature maps before and after Change-Enhanced Module.**

**Qualitative Results.** From Fig. 5, it's evident that change areas in our dataset exhibit complex and irregular shapes. Overall, our method produces more satisfactory visual results, effectively handling both large-scale complete changes and intricate structures. In Fig. 5(a), compared to other methods, we mitigate the impact of non-mining areas in the top left corner, reducing false positive predictions. Our approach achieves this by utilizing Dynamic Change-Aware Attention during the encoding stage to extract relevant change area features, as depicted in Fig. 5(c). Unlike most comparative methods struggling to capture complete large-scale changes due to insufficient global and local information, our method maintains internal compactness by integrating multiscale features at various stages, ensuring the integrity of detection areas.

## 5.3 Experiments on LEVIR and WHU Dataset.

**Quantitative Results.** Tab. 3 presents a comparison of change detection methods on the LEVIR [4] and WHU [12] datasets. Our approach achieves the highest F1 and IoU, with 93.76%, 88.25%, and 93.73%, 88.21% respectively. Among the comparative methods, BIT, SARAS, ChangeFormer and USSFCNet also demonstrate good performance, with USSFCNet and SARAS following closely with F1 of 93.13% and 89.90%. Our method significantly improves detection

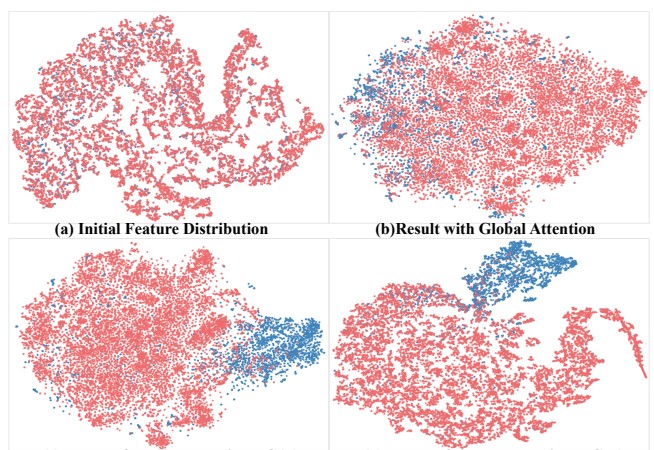

(a) Initial Feature Distribution  (b)Result with Global Attention

(c) Result after Incorporating DCAA  (d) Result after Incorporating MCEA

**Figure 8: Visualization of features enhanced by different modules via t-SNE.**

**Table 4: Validation of single-stream framework strategy. DMCA is a reimplementation of the dual-stream framework based on Multi-scale Change-Aware Transformer (MACT).**

| Models | F1 | IoU | Pre. | FLOPs (G) | #Param. (M) |
|---|---|---|---|---|---|
| MACT | **93.75** | **88.76** | **95.19** | 10.64 | 34.94 |
| DMCA | 92.67 | 87.64 | 93.34 | 16.83 | 38.35 |

accuracy by specifically targeting feature extraction in change areas, particularly in identifying complex and irregular changes.

**Qualitative Results.** Sample instances in Fig. 6 illustrate our method's excellent performance in detecting complex, large-area and small-area changes. In dense urban areas (as shown in Fig. 6(a) and (b)), our method accurately identifies changes and reduces noise, outperforming other methods especially in edge detection of buildings. This advantage stems from the multi-scale strategy that integrates features at various levels and preserves edge details. In

**Table 5: Ablations on the Dynamic Change-Aware Attention.**

| Models | F1 | IoU | Pre. | FLOPs (G) | #Param. (M) |
|---|---|---|---|---|---|
| MACT (Ours) | **93.75** | **88.76** | **95.19** | 10.64 | 34.94 |
| GA-Transformer | 93.21 | 88.67 | 95.16 | 11.72 | 35.02 |
| SA-Transformer | 92.57 | 88.48 | 95.13 | **10.48** | 32.71 |
| CA-Transformer | 93.51 | 88.37 | 95.16 | 10.72 | **34.25** |

**Table 6: Multi-scale Change-Enhanced Aggregator Ablation Study. Baseline indicates the absence of the aggregator, CEM denotes the Change-Enhanced Module, and MCA stands for Multi-scale Change Aggregator.**

| Baseline | CEM | MCA | F1 | IoU | Rec. |
|---|---|---|---|---|---|
| ✓ | ✗ | ✗ | 92.53 | 86.46 | 90.55 |
| ✓ | ✓ | ✗ | 92.96 | 87.51 | 91.31 |
| ✓ | ✗ | ✓ | 93.14 | 87.80 | 91.61 |
| ✓ | ✓ | ✓ | **93.75** | **88.76** | **92.40** |

terms of integrity preservation (as shown in Fig. 6(c)), our method effectively handles change areas of different scales, reducing prediction errors. When facing color-similar non-building areas (as shown in Fig. 6(d)), our method effectively excludes irrelevant information by interacting with change areas, enhancing detection accuracy.
**Visualization of Intermediate Features.** Fig. 7 demonstrates the substantial enhancement in feature extraction brought about by the Change-Enhanced Module. Prior to processing with this module, the feature maps show a vague delineation between change areas and background. In contrast, post-processing with the Change-Enhanced Module significantly sharpens the focus on these change areas, effectively distinguishing them from unchanged regions.

Additionally, we perform a two-dimensional projection of test set data points using t-distributed stochastic neighbor embedding (t-SNE). As shown in Fig. 8(a), the initial features display a mixed state of changed and unchanged pixels. The method using direct global attention (Fig. 8(b)) fails to effectively differentiate between the two types of pixels. In contrast, our Dynamic Change-Aware Attention (Fig. 8(c)) causes samples of the same class to cluster more tightly together. Finally, Fig. 8(d) indicates that the Multi-scale Change-Enhanced Aggregator successfully strengthens inter-class differences, eliminating the interference of irrelevant information.

In summary, our method consistently demonstrates strong performance. The Dynamic Change-Aware Attention and Multi-scale Change-Enhanced Aggregator encourage the model to effectively leverage and hierarchically aggregate local change features.

## 5.4 Ablation Study

To thoroughly validate the advantages of Multi-scale Change-Aware Transformer, a series of ablation studies are conducted. For details on model parameter selection, refer to the *Supplementary Materials*.
**Effect of Single-stream Pipeline.** We conduct experiments with a comparative analysis of frameworks to validate the superiority of the single-stream architecture. By streamlining the backbone and head structures, we juxtapose our approach against a dual-stream counterpart. Specifically, we develop a pyramidal structure,

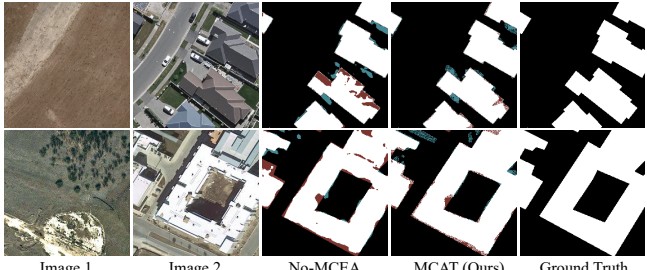

Image 1    Image 2    No-MCEA    MCAT (Ours)    Ground Truth

**Figure 9: Examples of results with and without Multiscal Aggregaator. The No-MCEA referring to our approach excludes the Multi-scale Change-Enhanced Aggregator.**

termed DMCA, which executes local attention operations at each stage and amalgamates the outputs from two encoders, all the while maintaining the integrity of the decoder structure. Tab. 4 delineates that while our revampe dual-stream model surpasses the performance of most advanced methods, it fails to match the finesse of our framework. This disparity underscores the efficacy of the single-stream architecture, whose integrated feature extraction facilitates seamless early-stage interaction between dual inputs.
**Analysis of Dynamic Change-Aware Attention.** To evaluate the effectiveness of the Dynamic Change-Aware Attention mechanism, we replace Dynamic Change-Aware Attention with other attention mechanisms, including Global Attention (GA), Self-Attention (SA), and Cross-Attention (CA). Data in Tab. 5 indicate that Dynamic Change-Aware Attention achieves the best overall performance under similar parameters and computational costs.
**Effect of Multi-scale Change-Enhanced Aggregator.** Tab. 6 shows that the introduction of the Change-Enhanced Module increased the F1 score and IoU score by 0.43% and 1.05% respectively. The model including the Multi-scale Change Aggregator shows significant improvements across all metrics compared to the baseline, especially with an increase of 0.61% in F1 and 1.34% in IoU. Fig. 9 also clearly highlight the significant performance improvements brought by Multi-scale Change Aggregator's multi-scale fusion strategy. The Multi-scale Change-Enhanced Aggregator enables the network to flexibly respond to changes in areas of various scales.

## 6 CONCLUSION

We introduce an innovative framework known as the Multiscale Change-Aware Transformer, which seamlessly integrates feature extraction and relationship modeling through a Dynamic Change-Aware Attention module, enhancing the feature extraction and interaction process between input image pairs. Furthermore, the Change-Enhanced Multiscale Aggregator targets the mining of multiscale features from change regions, significantly boosting the model's detection capabilities across various scales of change. In addition, we develop the Mining Area Change Detection dataset that encompasses complex morphologies and large-scale changes, effectively addressing the limitations of existing datasets that predominantly focus on changes with regular shapes. Our experimental results substantiate that our proposed method achieves state-of-the-art performance levels in change detection tasks, particularly excelling in the handling of complex and large-scale changes.

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
