# OpenReview forum: "Multi-scale Change-Aware Transformer for Remote Sensing Image Change Detection"
_acmmm.org/ACMMM/2024/Conference — MM2024 Poster_

### Official Review · Reviewer_VZCJ · 2024-04-29

**Rating:** 4
**Confidence:** 2

**Summary:**

This paper aims at solving change detection problems. Compared with previous methods, the authors introduce a single-stream architecture, the Multi-scale Change-Aware Transformer (MACT). It contains Dynamic Change-Aware Attention module and 	Multi-scale Change-Enhanced Aggregator to model the feature co-relation and mutli-scale detailed information.
Extensive experiments on various change detection benchmarks show the effectiveness of proposed method.

**Strengths:**

1, The proposed joint feature extraction and integration looks good and interesting to me.

2, The overall writing is good and easy to follow.

3, The proposed method MACT achieves stronger results on various benchmarks.

4, The proposed Dynamic Change-Aware Attention is interesting.

**Limitations:**

1, The novelties of Change-Enhanced Module (CEM) and Multi-scale Change Aggregator (MCA) are limited since both modules involve common operations including depth-wise convolution.
Moreover, multi-scale feature fusion is nothing but common operation in many fields, including semantic segmentation and object detection. The authors only apply the fusion operation for change detection.

2, Missing one important comparison studies to indicates the effectiveness of joint feature extraction.

3, Scope: I double this submission may not be suitable for ACM-MM. Maybe TGRS is better.

I am not expert on this topic. I list the both strength and weakness and give the boardline rating for reference.

**Suitability:**

2

---

### Official Review · Reviewer_8wcC · 2024-05-20

**Rating:** 4
**Confidence:** 3

**Summary:**

The paper discusses the challenges in remote sensing image change detection (CD) and introduces a new multi-scale change-aware transformer architecture called MACT. The proposed method presents a single-stream framework that contains a dynamic change-aware attention module and a multi-scale change-enhanced aggregator. MACT outperforms existing CD methods on the proposed dataset, MACD, and two popular benchmark datasets, LEVIR-CD and WHU-CD.

**Strengths:**

1. The motivation for this work is clearly expressed.
2. The figures of the proposed method are well-drawn and easy to understand.

**Limitations:**

1. The main issue of this work is that the details of the proposed multi-scale change-aware transformer are not shown in total, while the necessary source codes are not provided for review. Is this Transformer architecture related to the existing popular Transformer methods, e.g., Swin[1] and PVT[2]? Does it need to be pre-trained on ImageNet? And how are the hyperparameters set in the network, e.g., patch size, N_i {1,2,3,4}?

2. Eq. (18) shows that there are also four stages of predicted change maps for the proposed method. This is not made explicit in Figure 2 and Section 3.3.

3. In Figure 2, ‘DWConv’ indicates the depth-wise separable convolution, not only the depth-wise convolution.

4. In Eq. (10), the meaning of ‘CAT’ is not given. Is it concatenation? Also, Eq. (15) should make it clear that ‘\odot’ is the Hadamard product.

5. Transformer-based change detection methods should include LSAT [3], FHD [4], TransUNetCD [5], etc., in addition to SwinSUNET and ChangeFormer, which are introduced in the manuscript.

6. In the supplementary material, Table 1 should also give FLOPs and Params.

[1] Swin transformer: Hierarchical vision transformer using shifted windows, ICCV, 2021.
[2] Pyramid vision transformer: A versatile backbone for dense prediction without convolutions, ICCV, 2021.
[3] Lightweight Structure-Aware Transformer Network for Remote Sensing Image Change Detection, IEEE GRSL, 2023.
[4] Feature Hierarchical Differentiation for Remote Sensing Image Change Detection, IEEE GRSL, 2022.
[5] TransUNetCD: A Hybrid Transformer Network for Change Detection in Optical Remote-Sensing Images, IEEE TGRS, 2022.

**Suitability:**

3

---

### Official Review · Reviewer_W7hY · 2024-05-20

**Rating:** 5
**Confidence:** 4

**Summary:**

In this work, the authors collect a mining area change detection dataset that contains multiple scales and complex changing shapes. Besides, to improve the model's ability to interactively extract change information from remote sensing images, the authors propose the Multi-scale Change-Aware Transformer (MACT), which features the Dynamic Change-Aware Attention module and the Multi-scale Change-Enhanced Aggregator, reaching a good performance with less computational resources.

**Strengths:**

1. The authors propose a novel change detection dataset containing complex changing areas, which can pose more challenges to current models.
2. This work introduces a single-branch encoder and proves that it outperforms its dual-branch counterpart. If this can be universally useful to all other change detection methods, people can reduce computation complexity and reach better performance at the same time.
3. The proposed method reaches good performances on the Mining Area, LIVER,  and WHU datasets.

**Limitations:**

1. The proposed datasets contain 2133 pairs of images with $128 \times 128$ resolution. This is smaller than other existing datasets such as SYSU, WHU, etc. It is possible that limited training data could lead to over-optimization problems more easily.
2. The authors use the ratio of the perimeter to the area of change regions as a complexity metric, however, is this the best metric to evaluate complexity? For example, if the region is very small but still complex, the metric can hardly reflect the complexity.

[TYPOs]
1. Line 135: multi-scale

**Suitability:**

2

---

### Official Review · Reviewer_RhKc · 2024-05-29

**Rating:** 5
**Confidence:** 4

**Summary:**

This manuscript  proposed a single-stream architecture,  Multi-scale Change-Aware Transformer (MACT) . It contains two main modules, i.e., Dynamic Change-Aware Attention module and Multi-scale Change-Enhanced Aggregator. The Dynamic Change-Aware Attention module, integrating local self-attention and cross-temporal attention, conducts dynamic iteration on images differencesn to extract  features of change areas. The Multi-scale Change-Enhanced Aggregator enables the model to adapt to various scales and complex shapes through local change enhancement and multiscale aggregation strategies.
It also constructs Mining Area Change Detection dataset, which introduces complex and varied change area morphologies.

**Strengths:**

The Mining Area Change Detection dataset is meaningful for change detection research. It offers a diverse array of change areas that span multiple scales and exhibit complex shapes.
The proposed Multi-scale Change-Aware Transformer (MACT) performs well on MACD, LEVIR and WHU dataset.

**Limitations:**

1. The writing needs to be  improved. There are many mistakes, for example, the sentence  "Changes are categorized into (less than 5%), medium (5% to 15%), and large (greater than 15%) areas relative to the image size, as illustrated in Fig. 4(b). " misses  the word "small".
2. The loss contains three parts. The ablation experiments should be conducted on it to verify the effect of  these three parts.
3. In figure 1,  the structure of (c)(i)  is not  consistent with  the structure MCEA. A line from M1 to CEM is missed.

**Suitability:**

2

---

### Meta-Review · Area_Chair_jaN3 · 2024-06-30

**Recommendation:** Accept (Poster)
**Confidence:** 5

**Metareview:**

The AC goes through the paper, rebuttal and review comments.  This paper got 4 acceptances. After discussion, all the reviewers acknowledge the contribution of the novel change detection dataset. Nevertheless, the technical contribution of the proposed Multi-scale Change-Aware Transformer (MACT) can be further improved. In conclusion, the AC recommends accepting this paper.